# Collapsing the Learning: Crafting Broadly Transferable Unlearnable Examples

## Abstract

The success of Artificial Intelligence (AI) can be largely attributed to the availability of high-quality data for constructing machine learning models. Recently, the importance of data in AI has been significantly emphasized, leading to concerns regarding the secure utilization of data, particularly in the context of unauthorized usage. To address data exploitation, data unlearning has been introduced as a method to render data unexploitable by generating unlearnable examples. However, existing unlearnable examples lack the necessary generalization for broad applicability. In this paper, we propose a novel data protection method that generates robust transferable unlearnable examples, ensuring their effectiveness across diverse network architectures, even under challenging adversarial training conditions. To the best of our knowledge, our approach is the first to generate transferable unlearnable examples by leveraging data collapse as a means to reduce the information contained in data. Moreover, we modify the conventional adversarial training process to ensure that our unlearnable examples maintain robust transferability, even when the targeted model undergoes adversarial training. Comprehensive experiments demonstrate that the unlearnable examples generated by our method exhibit superior robust transferability compared to other state-of-the-art techniques.

## 1 Introduction

Over the last decade, the field of Artificial Intelligence (AI) has experienced significant advancements, leading to a substantial impact on nearly every domain. One of the crucial factors contributing to this success has been the access to an abundance of high-quality data for training machine learning models. Lately, the importance of data in AI has been greatly emphasized. Many major AI breakthroughs in natural language processing (Devlin et al., 2019; OpenAI, 2023), computer vision (Rombach et al., 2022; Oppenlaender, 2022) and computational biology (Jumper et al., 2021) have been realized only after obtaining the appropriate training data. As the role of data in artificial intelligence has been significantly magnified, concerns have arisen regarding the secure utilization of data. On one hand, some technology companies utilize unauthorized private data to train their commercial models (Vermes, 2023; Brittain, 2023; Growcoot, 2022); on the other hand, there are enterprises that wish to protect their data assets, ensuring that they are not exploited by competitors for model training, thereby maintaining a leading advantage in their own models (Fowl et al., 2021b).

To meet various data protection needs, recent research indicates that by injecting imperceptible noise into data, the performance of models employing such poisoned data can be significantly

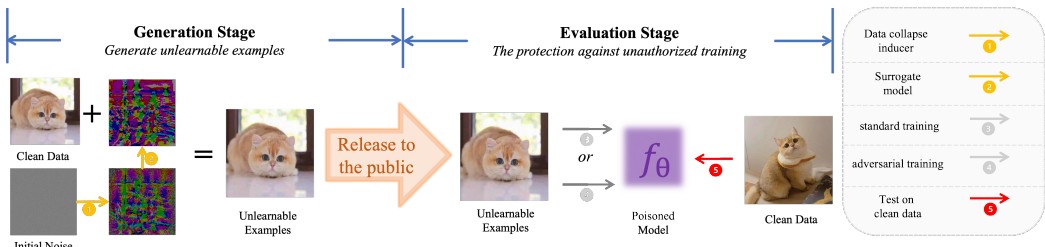

Figure 1: The generation and evaluation stages of unlearnable examples.

impaired (Fowl et al., 2021a;c; Yuan & Wu, 2021; Huang et al., 2021; Yu et al., 2022; Sadasivan et al., 2023; Zhang et al., 2023). The imperceptible noise is called unlearnable noise, while the poisoned data is also referred to as unlearnable data or unlearnable examples. Figure 1 shows the generation and evaluation stages of unlearnable examples. There are numerous methods to protect data from unauthorized use, such as LSP (Yu et al., 2022), EM (Huang et al., 2021), AR (Segura et al., 2022), TAP (Fowl et al., 2021c), NTGA (Yuan & Wu, 2021), CUDA (Sadasivan et al., 2023), and TUE (Ren et al., 2023). However, these methods are vulnerable to adversarial training in the evaluation stage. Once the targeted model undergoes adversarial training on these unlearnable examples (UEs), the protective effects of UEs are lost, and the model can still perform well on clean data. To maintain the protective effects of UEs under adversarial training, recent research has proposed REM (Fu et al., 2022) and EntF (Wen et al., 2023). While both of these methods offer protection under adversarial training, they do not explicitly consider *transferability, which refers to the protective effects of UEs generated by distinct surrogate models when evaluated on various target models.*

In practical scenarios, we hope that unlearnable examples generated by different surrogate models provide protective effects across various targeted model architectures, which is why we are more concerned about the issue of transferability. In this paper, we investigate unlearnable examples from the perspective of data distribution, aiming to enhance the transferability of unlearnable examples through the data itself. Specifically, we induce data collapse by utilizing the probability distribution of the data. In this approach, the data becomes more centralized, and the variation between different samples is effectively reduced, thereby diminishing the amount of useful information that can be learned by the model. Moreover, altering the data distribution is independent of any surrogate model, ensuring that our method exhibits high transferability.

To further advance our method's robust transferability, we propose a modified adversarial training process tailored to the task of unlearnable examples generation. Here, robust transferability refers to our unlearnable examples' ability to maintain effective unlearning protection across different network architectures during the evaluation phase, even under challenging adversarial training conditions. The following Figure 2 clearly displays the protective effects of the unlearnable examples generated by our method on different models across various datasets. We denote the test accuracy of models trained on clean data as $Acc_c$ and the test accuracy of

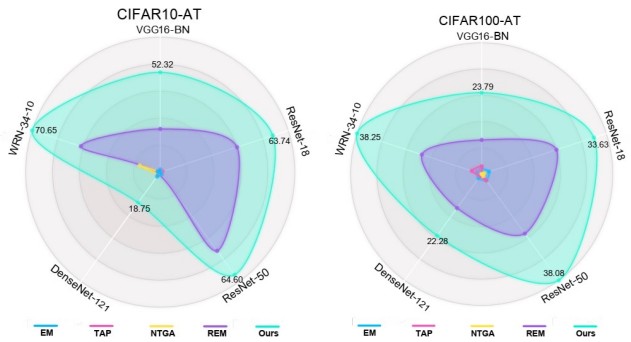

Figure 2: Protective effects of five type unlearnable methods on CIFAR-10 and CIFAR-100 with different adversarially trained target models.

models trained on unlearnable examples as $Acc_u$. Typically, $Acc_c$ is greater than $Acc_u$. Consequently, the protective effects of unlearnable examples can be defined as the difference between these accuracies, represented by $Acc_c - Acc_u$. A larger value indicates better performance. We can clearly observe a significant improvement in transferability compared to existing methods.

In summary, our main contributions are as follows:

- To the best of our knowledge, we are the first to generate unlearnable examples by inducing data collapse, which can not only reduce the information contained in the data but also improve the transferability of unlearnable examples.

- We propose a refined adversarial training process specifically tailored for the generation of unlearnable examples with robust transferability. This modified process ensures that unlearnable examples maintain effective unlearning protection and exhibit high transferability, even under demanding adversarial training conditions.

- We evaluate our method on CIFAR-10, CIFAR-100, and ImageNet-Subset with different categories and resolutions. Extensive experiments have verified our method outperforms other unlearnable methods both in protective effects and transferability.

## 2 RELATED WORK

**Non-robust unlearnable methods.** Methods of this nature typically generate non-robust unlearnable examples, implying that these examples can be readily compromised by adversarial training and consequently lose their protective effects. Notable instances of such methods include LSP (Yu et al., 2022), TUE (Ren et al., 2023), CUDA (Sadasivan et al., 2023), AR (Segura et al., 2022), TAP (Fowl et al., 2021c), NTGA (Yuan & Wu, 2021), and EM (Huang et al., 2021).

Methods such as LSP (Yu et al., 2022), AR (Segura et al., 2022), TUE (Ren et al., 2023), and CUDA (Sadasivan et al., 2023) generate unlearnable noise directly at the pixel level, devoid of any data information. Conversely, TAP (Fowl et al., 2021c), NTGA (Yuan & Wu, 2021), and EM (Huang et al., 2021) produce unlearnable noise by standardly training a surrogate model. This constraint limits the unlearnable noise, rendering it capable of protecting data only against standard training. Recent research (Fu et al., 2022) has demonstrated that these non-robust unlearnable methods are highly susceptible to adversarial training.

**Robust unlearnable methods.** Methods of this nature involve training robust surrogate models that learn robust features, such as REM (Fu et al., 2022) and EntF (Wen et al., 2023). REM (Fu et al., 2022) generates unlearnable noise for adversarial examples, as it posits that models can still acquire knowledge from adversarial examples. EntF (Wen et al., 2023) employs a pre-trained robust surrogate model to produce robust unlearnable examples.

Although these methods can provide protection under both standard training and adversarial training conditions, they do not address the issue of transferability. Furthermore, their focus is solely on protection from the perspective of training loss. Consequently, we propose enhancing protection by considering the data distribution and employing data collapse to improve the transferability of our approach. In contrast to these methods, we have adapted our approach to the unique characteristics of unlearnable examples by modifying the conventional adversarial training process. Through this modified adversarial training process, our unlearnable examples exhibit robust transferability.

## 3 METHODOLOGY

### 3.1 NOTATION

In this study, we address the problem of image classification using deep neural networks (DNNs) in the context of a $K$-class classification task. We define the clean training dataset as $\mathcal{D}_c$ and the test dataset as $\mathcal{D}_t$. Furthermore, we represent the classification DNN, which is trained on the clean dataset $\mathcal{D}_c$, by $f_\theta$, where $\theta$ denotes the parameters of the network. The primary objective of this research is to devise a method for transforming the training dataset $\mathcal{D}_c$ into an unlearnable dataset $\mathcal{D}_u$, such that any DNN trained on $\mathcal{D}_u$ exhibits poor performance when tested on the dataset $\mathcal{D}_t$.

Consider an original clean dataset $\mathcal{D}_c = \{(x_1, y_1), (x_2, y_2), \ldots, (x_n, y_n)\}$, consisting of $n$ clean samples, where $x_i \in \mathcal{X}$ denotes the $i$-th sample and $y_i \in \mathcal{Y} = \{1, \ldots, K\}$ represents the corresponding label. We define a surrogate model as $f'_\theta : \mathcal{X} \to \mathcal{Y}$, with $\theta \in \Omega$ signifying the model parameter. Furthermore, we introduce $\ell$ as a loss function. We also consider three types of perturbations: data collapse perturbation, denoted by $\delta_d$, adversarial perturbation, denoted by $\delta_a$, and unlearnable perturbation, denoted by $\delta_u$. Correspondingly, we define the data collapse perturbation radius as $\rho_d$, adversarial perturbation radius as $\rho_a$ and the unlearnable perturbation radius as $\rho_u$.

### 3.2 MOTIVATION

In real-world applications, it is desirable for unlearnable examples (UEs) generated by distinct surrogate models to be effective against a wide range of victim models. However, as discussed in Section 2, UEs created using non-robust methods are susceptible to being undermined by adversarial training. While current robustness approaches can be effective under both standard and adversarial training conditions, they do not adequately address the transferability of UEs.

In view of the aforementioned challenges, our primary goal is to develop a method for generating robust and transferable unlearnable examples. We approach this issue from two perspectives. Firstly, transferability is closely related to the underlying data; hence, we aim to enhance the transferability

of unlearnable examples by taking into account the data distribution. To achieve this, we initially train a data distribution gradient estimator, which is employed to modify the distribution of the data.

Subsequently, this estimator is utilized to induce data collapse, causing the data points to cluster together. Consequently, the differences between data points diminish, effectively concealing a portion of the data information. Obviously, modifying the data distribution from the perspective of the data itself is independent of the surrogate models and features, thereby exhibiting superior transferability. Secondly, To reinforce the robustness of transferability, we propose a modified adversarial training process for training the surrogate models. This process is designed to ensure that unlearnable examples maintain effective data protection, even under demanding adversarial training (Madry et al., 2018) conditions.

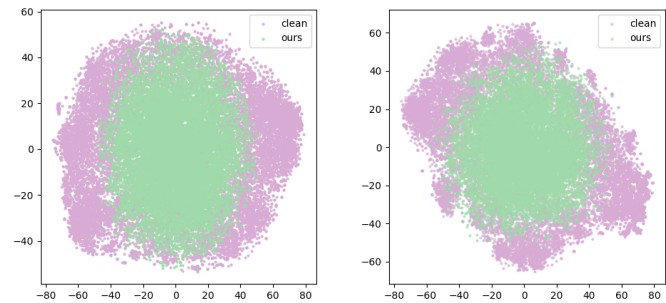

Figure 3: t-SNE visualizations of CIFAR-10 (left) and CIFAR-100 (right). We illustrate the t-SNE plots for clean (purple) examples, unlearnable examples generated by our method (green).

We contend that only robust surrogate models can generate unlearnable examples that are capable of withstanding the negative impacts of adversarial training. Figure 3 offers a persuasive visual illustration of the successful alteration of the data distribution achieved by inducing data collapse. This process effectively conceals the inherent knowledge contained within the data.

## 3.3 TRAINING DATA COLLAPSE INDUCER

Data collapse refers to the process of making data more concentrated, thereby reducing the amount of useful information that can be learned by a model. From the perspective of data distribution, this essentially involves altering the data distribution. We define the original data distribution as $p_D$, the collapsed distribution as $p'_D$, and the distribution transformation function as $\mathcal{T} : p_D \rightarrow p'_D$. By employing this distribution transformation function, we can induce data collapse, which in turn facilitates data unlearning from the perspective of data distribution. There are various methods for transforming data distribution. In this paper, we introduce the score matching technique to induce data collapse, owing to the widespread and mature application of diffusion models in the current research landscape. Let $x_t$ represent the sample at step $t$; $\nabla \log p_D(x_t)$ denote the log gradient of the data distribution concerning the sample $x_t$; $\alpha$ signify the step size; $\epsilon$ represents random noise following a distribution. $s_\theta$ indicate the data distribution gradient estimator, while $\tilde{x}$ refer to the sample after adding noise. According to Stochastic Gradient Langevin Dynamics (SGLD) (Zhu et al., 2021; Welling & Teh, 2011), we can iteratively update the original sample by:

$$x_t = x_{t-1} + \alpha \cdot \nabla_{x_{t-1}} \log p_D(x_{t-1}) + \sqrt{2\alpha} \cdot \epsilon. \tag{1}$$

As $\alpha \rightarrow 0$ and $t \rightarrow \infty$, $x_t$ converges to a sample from $p_D(x)$. In pursuit of data collapse, we eliminate the random noise term $\sqrt{2\alpha} \cdot \epsilon$ present in Equation 1 within our proposed method. Drawing inspiration from this function, we can readily manipulate the data distribution once we obtain the gradient of the ground truth data distribution.

In this section, we propose matching the gradient of the log ground truth data distribution, denoted by $s_\theta(x) = \nabla_x \log p_\theta(x)$. To accomplish this, we require training an estimator for the gradient of the log ground truth data distribution. The optimization objective is formulated as follows:

$$\frac{1}{2}\mathbb{E}_{p_D}[\|s_\theta(x) - \nabla_x \log p_D(x)\|_2^2]. \tag{2}$$

However, since the real data distribution is unknown, we require some preprocessing to transform it into a distribution that we can explicitly comprehend. Our approach involves adding Gaussian noise to the original data, thereby altering the data distribution $p_D$ into our predefined distribution. Let the training sample be denoted by $x$, the noise-added sample by $\tilde{x}$, and the predefined distribution

by $q_\sigma(\tilde{x}|x) \sim N\left(\tilde{x}; x, \sigma^2 I\right)$. Subsequently, the distribution of data after noise addition can be expressed as $q_\sigma(\tilde{x}) \equiv \int q_\sigma(\tilde{x}|x) p_D(x) dx$. By employing this method, the training objective can be reformulated as follows:

$$\frac{1}{2} \mathbb{E}_{q_\sigma(\tilde{x}|x) p_D(x)} \left[ \|s_\theta(\tilde{x}) - \nabla_{\tilde{x}} \log q_\sigma(\tilde{x}|x)\|_2^2 \right], \tag{3}$$

where $\nabla_{\tilde{x}} \log q_\sigma(\tilde{x}|x) = -\frac{\tilde{x}-x}{\sigma^2}$. Then, the final optimization object is as follows:

$$\frac{1}{2} \mathbb{E}_{q_\sigma(\tilde{x}|x) p_D(x)} \left[ \left\| s_\theta(\tilde{x}) + \frac{\tilde{x}-x}{\sigma^2} \right\|_2^2 \right]. \tag{4}$$

In this approach, we sample random noise, denoted by $\epsilon$, from the standard Gaussian distribution $N(0, I)$. Subsequently, we multiply it by our predefined $\sigma$ and add it to the sample $x$. As a result, this produces the noisy sample $\tilde{x} = x + \sigma\epsilon$. and the noise-added distribution $q_\sigma(\tilde{x}|x)$ satisfies $N\left(\tilde{x}; x, \sigma^2 I\right)$. By employing this method, we train an independent log gradient estimator, $s_\theta$, of the gradient of the log ground truth data distribution $\nabla_x \log p_\theta(x)$.

### 3.4 ROBUST TRANSFERABLE UNLEARNABLE EXAMPLES

By employing the log gradient estimator $s_\theta$, we can manipulate the data distribution to effectively conceal the inherent information. Instead of guiding the model to learn misinformation, we steer the data towards the gradient, ultimately inducing data collapse. In an ideal scenario, similar data points collapse at the same location, thereby minimizing variability between analogous data. As this approach focuses on alterations in the data distribution itself, it demonstrates superior transferability. The specific processing procedure can be outlined as follows:

$$x_i^t = x_i^{t-1} + \alpha \cdot s_\theta(x_i^{t-1}), \tag{5}$$

where $\alpha$ represents the step size. Then the data collapse perturbation is defined as $\delta_i^d = x_i^t - x_i^0$.

---

**Algorithm 1** Training our unlearnable examples' generator

---

**Input:** Training data set $\mathcal{D}$, training iteration $M$, score estimator $s_\theta$, learning rate $\eta$,
  1: Data collapse parameters $\rho_d$, $\alpha_d$ and $K_d$,
  2: PGD parameters $\rho_u$, $\alpha_u$, $\alpha_s$ and $K_u$,
  3: PGD parameters $\rho_a$, $\alpha_a$ and $K_a$.
**Output:** unlearnable examples generator $f_\theta'$.
  4: Initialize source model parameter $\theta$.
  5: **for** $i$ **in** $1, \cdots, M$ **do**
  6:      Sample a minibatch $(x, y) \sim \mathcal{D}$.
  7:      Initialize $\delta^d$.
  8:      **for** $s$ **in** $1, \cdots, K_d$ **do**                                          ▷ data collapse
  9:          $x \leftarrow x + \alpha_d \cdot s_\theta(x)$
10:          $\delta^d \leftarrow \delta^d + \alpha_d \cdot s_\theta(x)$
11:      **end for**
12:      $\delta^u = \delta^d$
13:      **for** $k$ **in** $1, \cdots, K_u$ **do**                 ▷ generate robust unlearnable examples
14:          $g_k \leftarrow \frac{\partial}{\partial \delta^u} \ell(f_\theta'(x + \delta^u), y)$
15:          $\delta^u \leftarrow \prod_{\|\delta\| \le \rho_u} (\delta^u - \alpha_u \cdot \text{sign}(g_k))$
16:      **end for**
17:      **for** $k$ **in** $1, \cdots, K_a$ **do**
18:          $g_k \leftarrow \frac{\partial}{\partial \delta^u} \ell(f_\theta'(x + \delta^u + \delta^a), y)$
19:          $\delta^a \leftarrow \prod_{\|\delta\| \le \rho_a} (\delta^a + \alpha_a \cdot \text{sign}(g_k))$
20:      **end for**
21:      $g_k \leftarrow \frac{\partial}{\partial \theta} \ell(f_\theta'(x + \delta^u + \delta^a), y)$
22:      $\theta \leftarrow \theta - \eta \cdot g_k$                                                  ▷ update $\theta$ of $f_\theta'$
23: **end for**
24: **return** $f_\theta'$

---

**Modified Adversarial Training.** As previously stated, our objective is to generate transferable unlearnable examples that exhibit robustness. Even when the targeted model undergoes adversarial training, our unlearnable examples can still offer effective data protection and maintain high transferability across diverse target model architectures. However, the generation process of unlearnable examples is rather distinctive, necessitating modifications to the conventional adversarial training in order to enhance the robustness of these unlearnable examples. As a result, we propose the following optimization objectives for adversarially training the surrogate model in our data unlearning task:

$$\min_{\theta} \frac{1}{n} \sum_{i=1}^{n} \max_{||\delta_i^a|| \leq \rho_a} \min_{||\delta_i^u|| \leq \rho_u} \ell(f'_\theta(x_i + \delta_i^u + \delta_i^a), y_i). \tag{6}$$

Introducing noise along the score function $s_\theta(x_i + \delta_i^d)$ can induce data collapse, effectively altering the data distribution. This process not only diminishes the differences between samples but also bolsters the transferability of unlearnable examples from the standpoint of the data itself. Subsequently, we assign $\delta_i^d$ to $\delta_i^u$, with the aim of enhancing the robustness of unlearnable examples for the collapsed distribution. After data collapse, we train the surrogate model by optimizing the aforementioned Equation 6. As for the modified adversarail training, the Projected Gradient Descent (PGD) (Madry et al., 2018) algorithm is firstly employed to modify $\delta_u$, which yields $x + \delta^u$. Subsequently, PGD is applied to the modified samples, $x + \delta^u$, in order to produce adversarial noise, thus obtaining $x + \delta^u + \delta^a$. Lastly, this resultant sample, $x + \delta^u + \delta^a$, is fed into the surrogate model to update $f'_\theta$. Details can be found in Algorithm 1. A robust transferable unlearnable example, denoted by $x'$, is constructed by adding the noise generated by the trained robust transferable noise generator $f'_\theta$ to its clean counterpart $x$. Consequently, the resulting unlearnable example is represented as $x' = x + \delta^u$.

## 4 EXPERIMENTS

### 4.1 EXPERIMENT SETUP

**Datasets.** To verify the effectiveness of our method on images of different categories and sizes, three commonly used datasets, namely CIFAR-10, CIFAR-100 (Krizhevsky et al., 2009), and ImageNet subset (Russakovsky et al., 2015) (consists of the first 100 classes), are used in our experiments. The data augmentation technique (Shorten & Khoshgoftaar, 2019) is adopted in every experiment.

**Models.** (1) As for the log gradient distribution estimator $s_\theta$, we employ U-Net (Ronneberger et al., 2015). (2) As for the surrogate model $f'_\theta$ in Eq.( 6), we employ VGG-16(Simonyan & Zisserman, 2015), ResNet-50(He et al., 2016), DenseNet-121(Huang et al., 2016), and ViT (Dosovitskiy et al., 2021). (3) VGG-16, ResNet-18, ResNet-50, DenseNet-121, ViT and Wide ResNet-34-10 (Zagoruyko & Komodakis, 2016) are used to evaluate the transferability of our method. (4) The $L_\infty$-bounded perturbation $\|\delta_u\|_\infty \leq \rho_u$ is adopted in our experiments.

**Metric.** We evaluate the data protection ability of unlearnable examples by measuring the test accuracy of the target models. A low test accuracy indicates that the model has learned little from the unlearnable examples, implying a strong protection ability.

### 4.2 EXPERIMENT RESULTS

#### 4.2.1 ROBUST EFFECTIVENESS OF OUR METHOD

To evaluate the robustness of our method against adversarial training, we initially introduce unlearnable noise to the entire training set, generating unlearnable examples. We train models using different adversarial training perturbation radii $\rho_a$ on these unlearnable examples, with $\rho_a$ ranging from $0/255$ to $4/255$. The unlearnable noise perturbation radius, denoted as $\rho_u$, is set to $8/255$ for all data unlearning methods, and the adversarial perturbation radius $\rho_a$ is set to $4/255$ for REM (Fu et al., 2022), EntF (Wen et al., 2023), and our method. The data collapse perturbation radius, denoted as $\rho_d$, is set to $8/255$. Table 1 presents the accuracies of the trained models on the unlearnable examples generated by different unlearnable methods.

The surrogate models in Table 1 are based on ResNet-18 (He et al., 2016). For adversarial training, we observe that even a minimal adversarial training perturbation radius of $2/255$ can undermine the protective effects of TAP (Fowl et al., 2021c), NTGA (Yuan & Wu, 2021), EM (Huang et al., 2021),

Table 1: Test accuracy (%) of models trained on unlearnable examples generated by different data unlearnable methods via adversarial training with different perturbation radii.

| Dataset | Adv. Train. $\rho_a$ | Clean | EM | TAP | NTGA | TUE | REM $\rho_a = 4/255$ | EntF $\rho_a = 4/255$ | Ours $\rho_a = 4/255$ |
|---|---|---|---|---|---|---|---|---|---|
| CIFAR-10 | 0 | 94.66 | 13.20 | 22.51 | 16.27 | 16.80 | 22.93 | 94.56 | **10.47** |
| | 1/255 | 93.74 | 22.08 | 92.16 | 41.53 | 92.58 | 30.00 | 93.56 | **14.33** |
| | 2/255 | 92.37 | 71.43 | 90.53 | 85.13 | 91.41 | 30.04 | 92.00 | **15.02** |
| | 3/255 | 90.90 | 87.71 | 89.55 | 89.41 | 90.51 | 31.75 | 91.04 | **17.33** |
| | 4/255 | 89.51 | 88.62 | 88.02 | 88.96 | 88.79 | 48.16 | 89.52 | **25.77** |
| CIFAR-100 | 0 | 76.27 | **1.60** | 13.75 | 3.22 | 1.71 | 11.63 | 75.83 | 2.70 |
| | 1/255 | 71.90 | 71.47 | 70.03 | 65.74 | 69.54 | 14.48 | 71.88 | **4.07** |
| | 2/255 | 68.91 | 68.49 | 66.91 | 66.53 | 67.66 | 16.60 | 68.94 | **9.83** |
| | 3/255 | 66.45 | 65.66 | 64.30 | 64.80 | 65.26 | 20.70 | 66.43 | **11.56** |
| | 4/255 | 64.50 | 63.43 | 62.39 | 62.44 | 63.33 | 27.35 | 63.94 | **17.35** |
| ImageNet Subset | 0 | 80.66 | **1.26** | 9.10 | 8.42 | 61.24 | 13.74 | 78.96 | 7.96 |
| | 1/255 | 76.20 | 74.88 | 75.14 | 63.28 | 63.43 | 21.58 | 75.34 | **13.20** |
| | 2/255 | 72.52 | 71.74 | 70.56 | 66.96 | 65.49 | 29.40 | 72.10 | **17.00** |
| | 3/255 | 69.68 | 66.90 | 67.64 | 65.98 | 64.89 | 35.76 | 67.88 | **21.42** |
| | 4/255 | 66.62 | 63.40 | 63.56 | 63.06 | 63.12 | 41.66 | 63.60 | **29.28** |

TUE (Ren et al., 2023), and EntF (Wen et al., 2023). However, our method exhibits generalizability to different perturbations while consistently maintaining effective protection. Even with a large perturbation of $4/255$, our method outperforms state-of-the-art techniques by $5\% \sim 19\%$. Our approach demonstrates effectiveness on three datasets, with varying numbers of categories and resolutions, indicating its applicability to images with different resolutions. Furthermore, as mentioned in Section 3.2, inducing data collapse can result in a more concentrated data distribution. We visualized the t-SNE plots of the CIFAR-10 and CIFAR-100 datasets before (purple) and after (green) protection in Figure 3. The t-SNE visualization reveals that the unlearnable examples generated by our method are more compact compared to the original data, effectively concealing sample differences. The experimental results further suggest that the unlearnable examples produced by our approach can efficiently reduce information, thereby diminishing the knowledge acquired by the model. These experiments and visualization demonstrate that our method can be applied to different datasets and adversarial training perturbation radii while maintaining substantial protective effects.

### 4.2.2 ROBUST TRANSFERABILITY OF OUR METHOD

Up until now, all our unlearnable examples have been generated using the ResNet-18 (He et al., 2016) surrogate model. As emphasized in Section 3.2, the enhancement of unlearnable examples' transferability is achieved through data collapse, indicating that it is not reliant on the choice of surrogate models. To further corroborate the transferability of our method, we aim to evaluate the protective effect of unlearnable examples generated by differnet surrogate models.

We investigate four surrogate models, including VGG-16, ResNet-18, ResNet-50, DenseNet-121, and ViT. Each type of unlearnable example is tested on five models, comprising VGG-16, ResNet-18, ResNet-50, DenseNet-121, WRN-34-10, and ViT. Experiments are conducted under both standard and adversarial training scenarios. The data collapse perturbation radius $\rho_d$ is set to $8/255$. The adversarial training perturbation radius is set to $4/255$, while the defensive perturbation radius $\rho_u$ for each unlearnable method is set to $8/255$.

Test accuracies of different standardly and adversarially trained target models on unlearnable CIFAR-10 examples generated by various surrogate models are presented in Table 2. Additional results on different datasets can be found in Appendix B. As depicted in Table 2, it is evident that our method can be applied to a variety of surrogate models while consistently achieving superior protective effects compared to other approaches. Notably, other methods exhibit sensitivity to the choice of surrogate models. In contrast, our method remains stable and effective, irrespective of the surrogate model's architecture. The absolute average accuracies of our unlearnable examples can reach $10\% \sim 30\%$ lower than other state-of-the-art methods.

In addition to the effectiveness and enhanced transferability under standard training conditions, our approach also attains remarkable transferability under adversarial training conditions. The unlearnable

Table 2: Test accuracy (%) of models standardly and adversarially trained on unlearnable examples generated by different surrogate models on CIFAR-10.

| Training Strategy | Surrogate Model | Method | VGG-16 | ResNet-18 | ResNet-50 | DenseNet-121 | WRN-34-10 | ViT | Average |
|---|---|---|---|---|---|---|---|---|---|
| Standard | VGG-16 | EM | 70.35 | 77.84 | 62.63 | 71.24 | 66.43 | 39.25 | 64.62 |
| | | REM | 46.27 | 45.25 | 40.29 | 43.29 | 42.95 | 32.82 | 41.81 |
| | | Ours | **26.59** | **26.23** | **29.41** | **32.98** | **30.25** | **22.82** | **28.05** |
| | ResNet-18 | EM | 15.70 | 13.20 | 10.19 | 14.59 | 10.56 | 72.62 | 22.81 |
| | | REM | 23.55 | 22.93 | 22.98 | 28.47 | 23.19 | 25.36 | 24.41 |
| | | Ours | **10.97** | **10.47** | **10.04** | **12.26** | **10.75** | **17.55** | **12.01** |
| | ResNet-50 | EM | 18.77 | 17.89 | 19.31 | 26.08 | 19.41 | **12.15** | 18.94 |
| | | REM | 24.64 | 23.22 | 23.91 | 30.04 | 22.82 | 35.57 | 26.37 |
| | | Ours | **10.01** | **11.23** | **11.51** | **11.91** | **10.41** | 18.38 | **12.24** |
| | DenseNet-121 | EM | 17.67 | 12.30 | 11.76 | 18.85 | 11.23 | 22.75 | 15.76 |
| | | REM | 35.46 | 35.91 | 33.09 | 33.26 | 33.49 | 33.73 | 34.16 |
| | | Ours | **16.14** | **11.30** | **11.10** | 18.38 | **10.91** | **21.68** | **14.92** |
| | ViT | EM | 30.35 | **29.66** | 31.63 | 70.85 | 23.83 | 67.61 | 42.32 |
| | | REM | 41.70 | 34.39 | 36.73 | 33.17 | 42.53 | 29.09 | 36.27 |
| | | Ours | **23.41** | 23.01 | **22.17** | **24.73** | **23.05** | **24.92** | **23.55** |
| Adversarial | VGG-16 | EM | 87.75 | 89.21 | 90.19 | 83.58 | 90.83 | 44.68 | 81.04 |
| | | REM | 73.60 | 74.73 | 74.16 | 77.63 | 74.94 | 46.52 | 70.26 |
| | | Ours | **64.72** | **54.17** | **62.35** | **70.21** | **62.36** | **29.01** | **57.14** |
| | ResNet-18 | EM | 87.24 | 88.62 | 89.66 | 81.77 | 90.96 | 46.94 | 80.87 |
| | | REM | 65.23 | 48.16 | 40.65 | 82.38 | 48.39 | 46.30 | 55.19 |
| | | Ours | **35.19** | **25.77** | **25.19** | **64.46** | **20.56** | **38.90** | **35.01** |
| | ResNet-50 | EM | 87.57 | 89.17 | 89.83 | 82.64 | 90.68 | 49.82 | 81.62 |
| | | REM | 51.88 | 44.27 | 37.79 | 82.01 | 42.09 | 55.72 | 52.29 |
| | | Ours | **42.64** | **30.04** | **28.78** | **74.42** | **30.44** | **47.09** | **42.24** |
| | DenseNet-121 | EM | 87.59 | 84.51 | 85.57 | 82.76 | 85.68 | 49.48 | 79.27 |
| | | REM | 67.30 | 69.62 | 66.42 | 60.51 | 72.09 | 48.61 | 64.09 |
| | | Ours | **35.88** | **36.61** | **31.39** | **39.35** | **34.78** | **41.40** | **36.57** |
| | ViT | EM | 87.40 | 89.12 | 89.59 | 82.51 | 90.54 | 51.96 | 81.85 |
| | | REM | 87.58 | 89.25 | 89.54 | 81.17 | 90.81 | 55.45 | 82.30 |
| | | Ours | **71.01** | **74.84** | **73.87** | **66.50** | **76.56** | **39.67** | **67.08** |

examples generated by surrogate models with diverse architectures exhibit superior protective effects on both standardly and adversarially trained target models, signifying that our method achieves robust transferability. The experimental results presented in Table 2 convincingly demonstrate that the combination of our proposed data collapse and the modified adversarial training process enables unlearnable examples to achieve robust transferability.

### 4.2.3 ABLATION STUDY

Through Sections 4.2.1 and 4.2.2, we have presented sufficient evidence that the unlearnable examples generated by our method exhibit robust transferability. As previously mentioned, we posited that data collapse could not only reduce the learnable knowledge contained within the data but also, owing to the inherent nature of data processing, result in unlearnable examples with strong transferability when employing data collapse techniques. Furthermore, to ensure that our generated unlearnable examples exhibit robust transferability, we modified the conventional adversarial training process when training the surrogate model. To further verify the individual effects of data collapse and the modified adversarial training process, we designed the following experiments. We employed ResNet-18 (He et al., 2016) as the surrogate model and separately removed the data collapse step and the modified adversarial training process to observe the test accuracy of the corresponding unlearnable examples on different target models. Target models are trained under standard or adversarial conditions. Table 3 presents the experimental results on CIFAR-10 and CIFAR-100.

In Table 3, **w/o(Collapse + Adv)** represents the removal of both data collapse and modified adversarial training, causing our method to degenerate into EM (Huang et al., 2021), denoted as $M_1$. Additionally, **w/o(Adv)** indicates the removal of the modified adversarial training, employing only data collapse, denoted as $M_2$. Meanwhile, **w/o(Collapse)** signifies the removal of the data collapse, incorporating modified adversarial solely during the surrogate model's training phase, denoted as $M_3$.

Compared to $M_1$, $M_2$ incorporates only the data collapse technique. We observe a significant improvement in transferability for the test accuracy of targeted models under standard training when

Table 3: Test accuracy (%) of targeted models trained on unlearnable examples generated by different ablation conditions. The surrogate model is ResNet-18.

| Dataset | Training Strategy | Training Method | VGG-16 | ResNet-18 | ResNet-50 | DenseNet-121 | WRN-34-10 |
|---|---|---|---|---|---|---|---|
| CIFAR-10 | Standard | w/o(Collapse + Adv) | 15.70 | 13.20 | 10.19 | 14.59 | 10.56 |
| | | w/o(Adv) | 11.20 | 11.44 | 10.72 | 12.50 | 10.23 |
| | | w/o(Collapse) | 16.68 | 20.19 | 14.28 | 17.93 | 13.53 |
| | | Ours | 10.97 | 10.47 | 10.04 | 12.26 | 10.75 |
| | Adversarial | w/o(Collapse + Adv) | 87.24 | 88.62 | 89.66 | 81.77 | 90.96 |
| | | w/o(Adv) | 87.21 | 89.31 | 89.74 | 82.26 | 89.74 |
| | | w/o(Collsape) | 54.25 | 37.40 | 36.42 | 77.61 | 36.36 |
| | | Ours | 35.19 | 25.77 | 25.19 | 64.46 | 20.56 |
| CIFAR-100 | Standard | w/o(Collapse + Adv) | 4.59 | 1.61 | 4.77 | 12.98 | 3.59 |
| | | w/o(Adv) | 1.89 | 2.11 | 2.20 | 2.63 | 2.18 |
| | | w/o(Collsape) | 5.42 | 8.41 | 8.55 | 9.31 | 10.03 |
| | | Ours | 4.38 | 2.70 | 3.30 | 7.84 | 2.50 |
| | Adversarial | w/o(Collapse + Adv) | 56.94 | 64.17 | 66.43 | 53.52 | 68.27 |
| | | w/o(Adv) | 57.45 | 64.43 | 66.38 | 53.89 | 67.82 |
| | | w/o(Collsape) | 59.24 | 22.83 | 22.81 | 52.50 | 20.41 |
| | | Ours | 55.12 | 17.35 | 17.26 | 52.12 | 15.39 |

comparing $M_2$ to $M_1$. This result validates that inducing data collapse not only reduces the learnable information contained in the data but also enhances the transferability of unlearnable examples. In contrast, $M_3$ introduces the modified adversarial training process during the surrogate model's training phase when compared to $M_1$. We observe a substantial decrease in test accuracy for targeted models under adversarial training when comparing $M_3$ to $M_1$. However, there is a slight increase in test accuracy for targeted models under standard training when comparing $M_3$ to $M_1$, which is a common phenomenon. By integrating data collapse and the modified adversarial training process, the unlearnable examples generated by our method are endowed with robust transferability.

### 4.2.4 Different protection percentages.

In a more realistic and challenging scenario, only a portion of the data is protected, while the remainder is clean. The detailed settings and results for this scenario can be found in Appendix C.

## 5 Conclusion and Limitation

**Conclusion.** In summary, in practical scenarios, we aim for unlearnable examples (UEs) generated by different surrogate models to provide protective effects across various target model architectures, which is why we place greater emphasis on the issue of transferability. In this paper, we approach this problem from the perspective of data collapse. We introduce data distribution gradients, which reflect the direction of concentrated areas in the data probability density. By inducing data collapse, we make the data more concentrated, thereby concealing the differences between data samples and reducing the knowledge that models can learn. Moreover, since this process alters the data distribution and involves manipulation of the data itself, it exhibits strong transferability. Through this approach, we can generate unlearnable examples with high transferability. Furthermore, we strive to enhance the robustness of the unlearnable examples generated by our method. However, the existing adversarial training process is unsuitable for this task. Consequently, we propose a modified adversarial training process to endow our unlearnable examples with robust transferability. A wealth of empirical evidence substantiates that only by incorporating data collapse and the modified adversarial training process can our method achieve superior data protection effects and robust transferability.

**Limitations.** The proposed method in this paper requires the introduction of a modified adversarial training process to generate robust transferable unlearnable examples. This leads to a significant computational cost when extending the method to large-scale datasets, such as ImageNet. As a result, the scalability of the proposed approach may be limited, especially when dealing with massive datasets. Furthermore, the method necessitates the additional training of a score model to represent the data distribution. Compared to other approaches, this requirement increases the generation cost of protective noise to a certain extent. Future research could explore ways to optimize the process or develop alternative techniques that can achieve similar results with lower computational overhead.

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

# A   DETAILED EXPERIMENTAL SETTINGS

## A.1   DATA AUGMENTATION

In our experiments, we employ distinct data augmentation techniques tailored to different datasets. For CIFAR-10 and CIFAR-100 (Krizhevsky et al., 2009), we apply data augmentation comprising random flipping, padding of 4 pixels on each side, random cropping to a size of $32 \times 32$, and rescaling per pixel in the range $[-0.5, 0.5]$ for individual images. In the case of the ImageNet subset, we implement data augmentation through random cropping, resizing to dimensions of $224 \times 224$, random flipping, and rescaling per pixel in the range $[-0.5, 0.5]$ for each image.

## A.2   ADVERSARIAL TRAINING

Adversarial training (Madry et al., 2018) is a commonly used method to improve model robustness. Standard adversarial training aims to solve the following min-max optimization problem:

$$\min_{\theta} \frac{1}{n} \sum_{i=1}^{n} \max_{||\delta_i|| \leq \rho_a} \ell(f_\theta(x_i + \delta_i), y_i) \tag{7}$$

## A.3   PROJECTED GRADIENT DESCENT

PGD (Madry et al., 2018) is a standard approach for solving inner maximization and minimization problems. It performs iterative projection updates to search for the optimal perturbation as follows:

$$\delta^{(k)} = \prod_{\|\delta\| \leq \rho} \left[ \delta^{(k-1)} + c \cdot \alpha \cdot \text{sign}\left( \frac{\partial}{\partial \delta} \ell(f_\theta(x + \delta^{(k-1)}), y) \right) \right] \tag{8}$$

where $k$ is the current iteration step ($K$ steps at all), $\delta^{(k)}$ is the perturbation found in the $k$-th iteration, $c \in \{-1, 1\}$ is a factor for controlling the gradient direction, $\alpha$ is the step size, and $\prod_{\|\delta\| \leq \rho}$ means the projection is calculated in the ball sphere $\{\delta : \|\delta\| \leq \rho\}$. The final output perturbation is $\delta^{(K)}$. Throughout this paper, the coefficient $c$ is set as 1 when solving maximization problems and $-1$ when solving minimization problems.

## A.4   DETAILED SETTINGS FOR OUR METHOD

**Models.** As for the log gradient distribution estimator, we employ U-Net (Ronneberger et al., 2015). Following (Huang et al., 2021) and (Fu et al., 2022), we employ ResNet-18(He et al., 2016) as the surrogate model $f'_\theta$ for trainging our unlearnable examples generator with Eq. (6). The $L_\infty$-bounded perturbation $\|\delta_u\|_\infty \leq \rho_u$ is adopted in our experiments. Additionally, we also use other surrogate models, including VGG-16(Simonyan & Zisserman, 2015), ResNet-50(He et al., 2016), DenseNet-121(Huang et al., 2016), and ViT (Dosovitskiy et al., 2021) to test the generalizability of our method.

**Unlearnable Examples' Evaluation.** Unlearnable examples generated by different methods are evaluated both on standard training and adversarial training (Madry et al., 2018). We focus on $L_\infty$-bounded noise $\|\rho_a\|_\infty \leq \rho_a$ in adversarial training. We conduct adversarial training on unlearnable examples with different target models, including VGG-16, ResNet-18, ResNet-50, DenseNet-121, and Wide ResNet-34-10 (Zagoruyko & Komodakis, 2016). including VGG-16 (Simonyan & Zisserman, 2015), ResNet-18, ResNet-50 (He et al., 2016), DenseNet-121 (Huang et al., 2016), and wide ResNet-34-10 (Zagoruyko & Komodakis, 2016). Note that when $\rho_a$ takes 0, the adversarial training degenerates to the standard training.

In our approach, we build upon REM (Fu et al., 2022) and utilize ResNet-18 (He et al., 2016) as the surrogate model $f'_\theta$. In our experiments, we employ $L_\infty$-bounded noise constraints, denoted as $\|\delta_u\|_\infty \leq \rho_u$ and $\|\delta_a\|_\infty \leq \rho_a$, where $\rho_u$ represents the unlearnable noise perturbation radius and $\rho_a$ denotes the adversarial perturbation radius. Both quantities may assume various values. The PGD settings are detailed in Table 4.

Regarding the CIFAR-10 and CIFAR-100 datasets, each surrogate model undergoes training via SGD over $5,000$ iterations, utilizing a batch size of $128$, a momentum factor of $0.9$, a weight decay factor of $0.0005$, an initial learning rate of $0.1$, and a learning rate scheduler that reduces the learning rate by a factor of $0.1$ every $2,000$ iteration.

Similarly, for the ImageNet subset, we train each surrogate model using SGD for $3,000$ iterations with a batch size of $128$, a momentum factor of $0.9$, a weight decay factor of $0.0005$, an initial learning rate of $0.1$, and a learning rate scheduler that decays the learning rate by a factor of $0.1$ every $1,200$ iteration.

Table 4: The settings of PGD (Madry et al., 2018) for the noise generations of error-minimizing noise (EM) (Huang et al., 2021), targeted adversarial poisoning noise (TAP) (Fowl et al., 2021c), neural tangent generalization attack noise (NTGA) (Yuan & Wu, 2021), robust error-minimizing noise (REM) (Fu et al., 2022), entangled features (EntF) (Wen et al., 2023) and our method in different experiments. $\rho_u$ denotes the defensive perturbation radius of different types of noise, while $\rho_a$ denotes the adversarial perturbation radius.

| Datasets | Noise Type | $\alpha_u$ | $K_u$ | $\alpha_a$ | $K_a$ | $\alpha_d$ | $K_d$ |
|---|---|---|---|---|---|---|---|
| CIFAR-10 CIFAR-100 | EM | $\rho_u/5$ | 10 | - | - | - | - |
| | TAP | $\rho_u/125$ | 250 | - | - | - | - |
| | NTGA | $\rho_u/10 \times 1.1$ | 10 | - | - | - | - |
| | REM | $\rho_u/5$ | 10 | $\rho_a/5$ | 10 | - | - |
| | EntF | $\rho_u/5$ | 10 | $\rho_a/5$ | 10 | - | - |
| | Ours | $\rho_u/5$ | 10 | $\rho_a/5$ | 10 | 0.000002 | 20 |
| ImageNet Subset | EM | $\rho_u/5$ | 7 | - | - | - | - |
| | TAP | $\rho_u/50$ | 100 | - | - | - | - |
| | NTGA | $\rho_u/8 \times 1.1$ | 8 | - | - | - | - |
| | REM | $\rho_u/4$ | 7 | $\rho_a/5$ | 10 | - | - |
| | EntF | $\rho_u/4$ | 7 | $\rho_a/5$ | 10 | - | - |
| | Ours | $\rho_u/4$ | 7 | $\rho_a/5$ | 10 | 0.000002 | 20 |

## A.5 MODEL TRAINING DETAILS

In accordance with Eq. (7), we carry out adversarial training (Madry et al., 2018) as described in appendix A.2. Analogous to the training of the unlearnable examples generator, we focus on the $L_\infty$-bounded noise, denoted as $\|\rho_a\|_\infty \leq \rho_a$, during the adversarial training process.

Throughout our experiment, the model undergoes training using SGD for $40,000$ iterations with a batch size of $128$, a momentum factor of $0.9$, a weight decay factor of $0.0005$, an initial learning rate of $0.1$, and a learning rate scheduler that reduces the learning rate by a factor of $0.1$ every $16,000$ iteration. For the CIFAR-10 and CIFAR-100 datasets, the step number $K_a$ and the step size $\alpha_a$ in PGD are set to $10$ and $\rho_a/5$, respectively.

For the ImageNet subset, we set the step number $K_a$ and step size $\alpha_a$ to $8$ and $\rho_a/4$, respectively. This ensures a consistent and well-structured experimental setup across various datasets.

## B ROBUST TRANSFERABILITY

As mentioned in Section 4.2.2, Table 5 shows the test accuracies of different standardly and adversarially trained targeted models on unlearnable CIFAR-100. We examine four surrogate models, including VGG-16 (Simonyan & Zisserman, 2015), ResNet-18(He et al., 2016), ResNet-50 (He et al., 2016), and DenseNet-121 (Huang et al., 2016). Each type of unlearnable example is tested on five models, comprising VGG-16, ResNet-18, ResNet-50, DenseNet-121, and WRN-34-10 (Zagoruyko & Komodakis, 2016). Experiments are conducted under both standard and adversarial training scenarios. The adversarial training perturbation radius is set to $4/255$, while the defensive perturbation radius $\rho_u$ for each unlearnable method is set to $8/255$.

The experimental results presented in Table 5 further substantiate the high robust transferability of our method. In terms of average accuracy, our approach demonstrates a reduction of $6\% \sim 15\%$ when compared to the state-of-the-art methods. In conjunction with Table 2, our method maintains superior protective effects even when applied to different datasets.

Table 5: Test accuracy (%) of models standardly and adversarially trained on unlearnable examples generated by different surrogate models on CIFAR-100. The surrogate model is ResNet-18.

| Training Strategy | Surrogate Model | Method | VGG-16 | ResNet-18 | ResNet-50 | DenseNet-121 | WRN-34-10 | Average |
|---|---|---|---|---|---|---|---|---|
| Standard | VGG-16 | EM | 17.45 | 21.79 | 28.94 | 61.85 | 34.50 | 32.91 |
| | | REM | 11.89 | 21.08 | 19.76 | 17.95 | 26.72 | 19.48 |
| | | Ours | **9.27** | **11.50** | **11.00** | **13.86** | **10.97** | **11.32** |
| | ResNet-18 | EM | 4.59 | **1.60** | 4.77 | 12.98 | 3.59 | 5.51 |
| | | REM | 9.15 | 11.63 | 9.23 | 13.06 | 11.74 | 10.97 |
| | | Ours | **4.38** | 2.70 | **3.30** | **7.84** | **2.50** | **4.14** |
| | ResNet-50 | EM | 69.10 | 74.79 | 74.16 | 65.18 | 76.71 | 71.99 |
| | | REM | 10.24 | 10.53 | 9.43 | 23.28 | 10.15 | 12.73 |
| | | Ours | **4.21** | **6.92** | **6.86** | **6.38** | **7.63** | **6.40** |
| | DenseNet-121 | EM | **1.00** | 7.84 | 7.53 | 64.76 | 11.15 | 18.46 |
| | | REM | 16.62 | 20.28 | 18.88 | 17.96 | 20.91 | 18.93 |
| | | Ours | 3.52 | **5.07** | **5.20** | **8.32** | **10.22** | **6.47** |
| Adversarial | VGG-16 | EM | 57.33 | 63.55 | 65.44 | 53.45 | 68.23 | 61.60 |
| | | REM | 41.13 | 52.00 | 51.77 | 48.92 | 56.05 | 49.97 |
| | | Ours | **34.02** | **46.66** | **46.55** | **41.93** | **49.23** | **43.68** |
| | ResNet-18 | EM | 56.94 | 64.17 | 66.43 | 53.52 | 68.27 | 61.87 |
| | | REM | 58.07 | 27.35 | 26.03 | 56.63 | 27.71 | 39.16 |
| | | Ours | **55.12** | **17.35** | **17.26** | **52.12** | **15.39** | **31.45** |
| | ResNet-50 | EM | 56.82 | 64.19 | 66.93 | 54.51 | 68.56 | 62.20 |
| | | REM | 54.61 | 35.50 | 30.43 | 54.26 | 35.11 | 41.98 |
| | | Ours | **51.24** | **26.01** | **21.81** | **52.32** | **22.89** | **34.85** |
| | DenseNet-121 | EM | 57.39 | 63.73 | 66.37 | 54.62 | 68.43 | 62.11 |
| | | REM | 47.22 | 41.89 | 45.49 | 41.15 | 50.66 | 45.28 |
| | | Ours | **33.67** | **29.80** | **28.85** | **31.45** | **30.39** | **30.83** |

## C  DIFFERENT PROTECTION PERCENTAGES.

In a more challenging and realistic scenario, one might encounter a situation where only a subset of the data is protected by unlearnable noise, while the remainder is clean. To simulate this scenario, we randomly select a portion of the training data from the complete training set and introduce unlearnable noise to this selected data. Subsequently, we conduct adversarial training using ResNet-18 on the combination of mixed data (containing both unlearnable and clean examples).

The unlearnable perturbation radius for each noise instance is established at $8/255$, while the adversarial perturbation radius $\rho_a$ for both REM (Fu et al., 2022), EntF (Wen et al., 2023), and our method is set at $4/255$. The discrepancy between test accuracies obtained using mixed data and clean data serves as an indicator of the knowledge acquired from the protected training data.

The reported accuracies on clean test data are presented in Table 6, showcasing the impact of our approach when dealing with mixed datasets containing both protected and clean data samples. Table 6 reveals that, across varying mixture percentages, our proposed method consistently achieves superior data protection.

As previously mentioned, we manipulate the data distribution to induce data collapse, thereby reducing information. The obtained results corroborate our hypothesis. Furthermore, this evidence indicates that the unlearnable examples generated by our approach can provide enhanced protection, even when combined with clean data.

### C.1  PROTECTIVE EFFECTS AGAINST OTHER PURIFYING METHODS.

Before our method was proposed, there were two methods that could be used to undermine the protective effect of existing unlearnable examples generation methods, namely ISS (Liu et al., 2023) and Ueraser (Qin et al., 2023). ISS exploits the low-frequency and high-frequency characteristics of noise by using Gray and JPEG compression processing, respectively, thus disrupting the protective effect of unlearnable examples. On the other hand, Ueraser (Qin et al., 2023) works by applying a certain number of data augmentation methods, finding the augmentation method that maximizes the error for the corresponding sample, and using this to disrupt the protective effect of unlearnable

Table 6: Test accuracy (%) on CIFAR-10 and CIFAR-100 with different protection percentages.

| Dataset | Adv. Train. $\rho_a$ | Noise Type | 0% | 20% Mixed | 20% Clean | 40% Mixed | 40% Clean | 60% Mixed | 60% Clean | 80% Mixed | 80% Clean | 100% |
|---|---|---|---|---|---|---|---|---|---|---|---|---|
| CIFAR-10 | 2/255 | EM | 92.37 | 92.33 | 91.30 | 92.18 | 90.31 | 92.00 | 88.65 | 92.06 | 83.37 | 71.43 |
| | | TAP | | 92.17 | | 91.62 | | 91.32 | | 91.48 | | 90.53 |
| | | NTGA | | 92.41 | | 92.19 | | 92.23 | | 91.74 | | 85.13 |
| | | REM | | 92.23 | | 90.79 | | 88.85 | | 83.70 | | 30.04 |
| | | Ours | | **92.16** | | **90.51** | | **88.40** | | **82.97** | | **18.80** |
| | 4/255 | EM | 89.51 | 89.39 | 88.17 | 89.09 | 86.76 | 89.41 | 85.07 | 89.41 | 79.41 | 88.62 |
| | | TAP | | 89.01 | | 88.66 | | 88.40 | | 88.04 | | 88.02 |
| | | NTGA | | 89.56 | | 89.35 | | 89.22 | | 89.17 | | 88.96 |
| | | REM | | 89.71 | | 89.89 | | 89.63 | | 87.17 | | 48.16 |
| | | Ours | | **88.89** | | **88.53** | | **87.97** | | **84.03** | | **29.20** |
| CIFAR-100 | 2/255 | EM | 68.91 | 68.68 | 66.54 | 68.80 | 64.21 | 68.28 | 58.35 | 68.70 | 47.99 | 68.49 |
| | | TAP | | 68.40 | | 67.93 | | 67.25 | | 67.09 | | 66.91 |
| | | NTGA | | 68.52 | | 68.82 | | 68.36 | | 68.71 | | 66.53 |
| | | REM | | 68.90 | | 68.29 | | 61.42 | | 51.99 | | 16.60 |
| | | Ours | | **66.16** | | **67.58** | | **60.04** | | **48.70** | | **5.21** |
| | 4/255 | EM | 64.50 | 64.65 | 61.73 | 63.82 | 57.61 | 64.19 | 53.86 | 64.32 | 44.79 | 63.43 |
| | | TAP | | 64.36 | | 63.35 | | 62.58 | | 63.15 | | 62.39 |
| | | NTGA | | 63.48 | | 63.59 | | 63.64 | | 62.83 | | 62.44 |
| | | REM | | 64.27 | | 64.67 | | 64.99 | | 63.14 | | 27.35 |
| | | Ours | | **63.36** | | **63.22** | | **62.47** | | **62.63** | | **22.35** |

Table 7: Test accuracy (%) of models trained with ISS and Ueraser on unlearnable examples generated by EM, REM, and our method.

| Dataset | Method | EM | REM | Ours |
|---|---|---|---|---|
| CIFAR-10 | ISS-Gray | 21.73 | 79.52 | **20.99** |
| | ISS-JPEG10 | 85.12 | 85.39 | **83.56** |
| | Ueraser | 53.17 | 70.28 | **29.64** |
| CIFAR-100 | ISS-Gray | 60.19 | 45.38 | **5.67** |
| | ISS-JPEG10 | 67.48 | 63.82 | **29.07** |
| | Ueraser | 72.08 | 55.65 | **25.28** |

examples. We have tested the protective effect of our method against ISS (Liu et al., 2023) and Ueraser (Qin et al., 2023).

Additionally, we have also tested the performance of our method against ISS and Ueraser on the ImageNet subset.

Clearly, Table 7 and Table 8 demonstrate that our method still provides superior protection against existing defense methods.

Table 8: Test accuracy (%) of models trained with ISS and Ueraser on unlearnable ImageNet-Subset generated by our method.

| ISS-Gray | ISS-JPEG10 | Ueraser |
|---|---|---|
| 24.12 | 12.21 | 10.35 |

