# OpenReview forum: "Collapsing the Learning: Crafting Broadly Transferable Unlearnable Examples"
_ICLR.cc/2024/Conference — Submitted to ICLR 2024_

### Official Review · Reviewer_x3eU · 2023-10-31

**Soundness:** 2 fair
**Presentation:** 1 poor
**Contribution:** 2 fair
**Rating:** 3
**Confidence:** 5

**Summary:**

The paper proposes a method to generated robust transferable unlearnable examples.

**Strengths:**

1. Generating effective unlearnable examples that are transferable are important task.

**Weaknesses:**

1. The writing of the paper is extremely poor. The paper is not well-organized and it is difficult to follow.
2. The paper does not motivate the proposed method. What is the need for transferable unlearnable examples? On that same note, the contributions of the paper and the proposed method are not well-justified. Why does the proposed method work?
2. How does the proposed modified adversarial training help with effectiveness of the generated examples for an adversarial trained model?

**Questions:**

Please refer to the weaknesses.

---

> ### Author Response · Authors · 2023-11-23
>
> Thanks for your comments and your appreciation of our findings. We address your concerns below:
> > W1: The writing of the paper is extremely poor. The paper is not well-organized and it is difficult to follow.
>
> Thank you for your comment regarding the writing quality and organization of our paper. We have taken your feedback into account and have thoroughly revised and improved the manuscript based on the suggestions from you and other reviewers. The latest version has been submitted for your consideration.
>
>
> > W2: The paper does not motivate the proposed method. What is the need for transferable unlearnable examples? On that same note, the contributions of the paper and the proposed method are not well-justified. Why does the proposed method work?
>
> Thank you for your question regarding the motivation and justification for our proposed method. In practical scenarios, it is desirable for unlearnable examples generated by different surrogate models to provide protective effects across various targeted model architectures. This is why we focus on the issue of transferability.
> To achieve this goal, we first propose to induce data collapse. Using the Score Function [1], we obtain the log gradient of the data distribution. By updating the data with this log gradient, we make the original data more concentrated, thereby reducing the information contained in the data. As this process is model-agnostic, it possesses high transferability. Furthermore, to maintain protection under adversarial training conditions, we propose a modified adversarial training strategy for unlearnable examples. Ultimately, our method can achieve robust transferability. The effectiveness of our approach is demonstrated through numerous experiments in the paper.
> We hope this response addresses your concerns about the motivation and justification for our proposed method.
>
>
> > W3:  How does the proposed modified adversarial training help with effectiveness of the generated examples for an adversarial trained model?
>
> Thank you for your question regarding the effectiveness of our modified adversarial training. The modified adversarial training process helps make the surrogate model robust. In cases where the poisoned model undergoes adversarial training, it will also become robust. As a result, by utilizing robust surrogate models, we aim to generate robust unlearnable examples that remain effective even for adversarially trained models.
>
>
> If you have any specific question, I would be very willing to address your question.
>
> - \[1\] A connection between score matching and denoising autoencoders. Neural computation 2011

---

> ### Author Response · Authors · 2023-11-23
>
> We are truly grateful for your  feedback. In our responses, we have endeavored to address every concern raised with thorough explanations and evidence. As we approach the conclusion of this phase of the review process, we are keen to ascertain whether our rebuttal has successfully resolved the issues highlighted.
> We invite any additional comments or questions you may have regarding our responses. Your expertise and perspectives are vital in guiding the refinement and understanding of our work. We eagerly await your further guidance and insight.

---

### Official Review · Reviewer_KXx5 · 2023-11-01

**Soundness:** 2 fair
**Presentation:** 3 good
**Contribution:** 2 fair
**Rating:** 5
**Confidence:** 2

**Summary:**

The paper proposes to generate unlearnable examples by essentially modifying the REM method by first learning a log-gradient estimator $s_\theta$, then use it to induce data collapse with perturbations, before the standard REM procedure. The authors hypothesized this can help improve effectiveness and transferability, and provided experiments aiming to justify this claim.

**Strengths:**

1. While it is relatively well known that out-of-distribution perturbations can lead to transferrable adversarial attacks, the paper is the first to apply the idea on unlearnable examples.
  2. Experiments demonstrate that the unlearnable examples generated by the proposed method exhibit superior robust transferability compared to EM and REM.

**Weaknesses:**

1. The perturbation budget of adversarial training ($\ell_p \leq 4/255$) is too low, larger perturbations need to be considered. Especially, unlearnable perturbations can often be much less effective by simply increasing the adversarial perturbation budget.
  2. This paper lacks a comparison of an important baseline OPS [1]. A significant advantage of OPS is its strong resistance against $\ell_{\{2, \infty\}}$ adversarial training, even under large adversarial perturbation budgets.
  3. The generation of poisons seems to have a high computational overhead, and requires prior training of $s_\theta$, which can be very time-consuming and limits the practicability of proposed method.
  4. Only adversarial training is considered as a defense method, lacks of evaluations of recently proposed defenses (ISS [2], UEraser [3], AVATAR [4]).
  5. Please also include transformer-based models in Table 2.

- [1] One-pixel shortcut: on the learning preference of deep neural networks. ICLR 2023
- [2] Image shortcut squeezing: Countering perturbative availability poisons with compression. ICML 2023
- [3] Learning the unlearnable: Adversarial augmentations suppress unlearnable example attacks. https://arxiv.org/abs/2303.15127
- [4] The Devil's Advocate: Shattering the Illusion of Unexploitable Data using Diffusion Models. https://arxiv.org/abs/2303.08500

**Questions:**

### Questions:
  1. Are there significant advantages of the proposed method over OPS?
  2. The proposed method seems to be only a minor modification on REM and requires even more computational overhead than REM.
  3. Is the effectiveness of the proposed method still guaranteed if higher budget are considered for adversarial training? Or is it like REM where the unlearnable effect is significantly reduced?
### Other issues:
  1. The design of the log-gradient estimator is unexplained.
  2. Strangely, the hyperparameters $\rho_d$, $\alpha_d$ and $K_d$ are not defined properly, and $\rho_d$ is never assigned value or used in the paper. Also, the update on $x$ is unconstrained.
  3. Citations are not properly parenthesized, i.e. `\cite{…}` -> `\citep{…}`.

---

> ### Author Response · Authors · 2023-11-23
>
> Thanks for your valuable comments and your appreciation of our findings.  We address your concerns below:
> > Summary: The paper proposes  to induce data collapse, before the standard REM procedure.
>
> We would like to clarify that our approach is notably different from the REM method. The key distinctions are as follows:
> 1. To enhance the transferability of unlearnable examples, we propose **Data Collapse**. By employing data collapse, we reduce the information contained in the data. The ablation experiments in Table 3 of the paper demonstrate that data collapse can indeed improve transferability, which is a feature not present in other related works.
> 2. To enhance the robustness of transferability, we propose Modified Adversarial Training (MAT). This method specifically modifies traditional adversarial training to address the unique characteristics of unlearnable examples.  The optimization objective of MAT is **(min-max)-min**, while the optimization objective of REM is **min-(min-max)**. **The two objectives are fundamentally different**. Our optimization objective aims to make the substitute model $f_{\theta}'$ robust, thereby making the generated unlearnable examples robust as well. In contrast, REM's goal is to generate unlearnable examples for adversarial samples, and its substitute model is not robust.
> 3. Our method achieves robust transferability through two steps: Data Collapse and Modified Adversarial Training.
> We hope this clarification addresses your concerns regarding the differences between our approach and the REM method.
>
> > W1: The perturbation budget of adversarial training $(ℓ≤4/255)$ is too low, larger perturbations need to be considered. Especially, unlearnable perturbations can often be much less effective by simply increasing the adversarial perturbation budget.
>
> Thank you for your comment regarding the perturbation budget. In response to your concern, we have increased both $\rho_u$ and $\rho_a$, and the experimental results are presented in the table below.
>
> | | $\rho_a = 0$ | $\rho_a=8/255$ | $\rho_a=16/255$ | $\rho_a=32/255$ | $\rho_a=64/255$ |
> | - | :-: | :-: | :-: | :-: | :-: |
> |Clean| 94.66 | 84.79 | 70.41 | 42.76 | 10.00
> |Ours($\rho_u=16/255$)| 10.21 | 15.80 | 71.73 | 43.75 | 10.00 |
> |Ours($\rho_u=32/255$) | 10.35 | 10.46 | 18.07 | 53.83 | 10.00 |
> |Ours($\rho_u=64/255$) | 10.15 | 10.27 | 11.01 | 54.30 | 10.00 |
>
> As you pointed out, increasing $\rho_a$$ does indeed reduce the effectiveness of unlearnable examples. To clarify, we define the party generating unlearnable examples as the Attacker and the party training models using unlearnable examples as the Defender. In classification tasks, the Attacker's goal is to minimize the test accuracy of the Defender's model trained on unlearnable examples, while the Defender's goal is to achieve higher model accuracy.
> 1. From the Attacker's perspective, increasing $\rho_u$ has little cost, as there is no need to use unlearnable examples. However, for the Defender, undermining the effect of unlearnable examples requires increasing $\rho_a$, which is costly. As seen in the table above, as $\rho_a$ increases, the performance upper bound of the model declines significantly. This is actually an arms race, where the Attacker forces the Defender to raise $\rho_a$ by increasing $\rho_u$. Once $\rho_a$ reaches a certain level, the model's training difficulty increases, and the performance upper bound is significantly reduced. In this way, the Attacker achieves their goal.
> 2. In this scenario, Table 1 in the paper indicates that our method is superior to other methods. our method provides valuable insights into the dynamics between the Attacker and Defender and showcases its effectiveness in various settings, rather than being limited to a specific perturbation budget.
> We hope this response addresses your concerns and provides a clearer understanding of our approach.
>
> > W2 : This paper lacks a comparison of OPS [1]. A significant advantage of OPS is its strong resistance against $ℓ_{2,∞}$ adversarial training, even under large adversarial perturbation budgets.
>
> Thank you for your comment regarding the comparison with the OPS baseline. We acknowledge the importance of including relevant baselines for a comprehensive evaluation. In practice, **data augmentation methods are often employed** during the model training process, and their cost is **significantly lower** than that of adversarial training. OPS is prone to being compromised by data augmentation techniques([2,3]). The augmentation techniques almost have no effects on our method.  The table below shows the protect effects on CIFAR10 with data enhancements.
>
> |  |w/o| Median | CutMix | Cutout | ISS-JPEG10 |
> | :-: | :-: | :-: | :-: | :-: | :-: |
> | OPS | 36.55 | 85.16 | 76.40 | 67.94| 82.53 |
> | Ours | 10.47 | 18.03 | 11.05 | 14.86 | 38.11|
>
> We hope this response addresses your concerns and provides a clearer understanding of our approach in comparison to the OPS baseline.

---

> > ### Author Response · Authors · 2023-11-23
> >
> > > W3:  The generation of poisons seems to have a high computational overhead, and requires prior training of $s_{\theta}$, which can be very time-consuming and limits the practicability of proposed method.
> >
> > Thank you for your comment regarding the computational overhead and the time-consuming nature of training . We would like to provide the following clarifications:
> > 1. Since the original data remains unchanged, training  requires only one iteration. This process takes approximately 8 hours of 1 V100 GPU time on CIFAR.
> > 2. In our paper, we specifically employ the score matching method to induce data collapse. DDPM is essentially consistent with score matching, allowing us to utilize open-source checkpoints.
> > 3. The table below presents a comparison of the overhead time between our method and existing methods such as EM, TAP, and REM:
> >
> > |Dataset|EM|TAP|REM|Ours|
> > |:-|:-:|:-:|:-:|:-:|
> > |CIFAR-10|0.4h|0.5h|22.6h| 8.6h |
> > |CIFAR-100|0.4h|0.5h|22.6h| 8.6h |
> > |ImageNet(subset)|3.9h|5.2h|51.2h| 26.5h|
> >
> > We hope these clarifications address your concerns about the computational overhead and practicability of our proposed method.
> >
> > > W4: Only adversarial training is considered as a defense method, lacks of evaluations of recently proposed defenses (ISS [2], UEraser [3], AVATAR [4]).
> >
> > Thank you for your suggestion to evaluate our method against recently proposed defenses, such as ISS [2], UEraser [3], and AVATAR [4]. We have conducted additional experiments on CIFAR and ImageNet-subset datasets and present the results in the tables below.
> > CIFAR：
> >
> > |Dataset |Defense Method| Ours| EM | REM |
> > | :--- | :--- | :---: | :---: | :----: |
> > |CIFAR-10| ISS-Gray | **20.99** | 21.73 | 70.52 |
> > |CIFAR-10| ISS-JPEG10 | **38.11** | 85.12 | 85.39 |
> > |CIFAR-10| Ueraser | **29.64** | 53.17 | 70.28 |
> > |CIFAR-100| ISS-Gray | **5.67** | 60.19 | 45.38 |
> > |CIFAR-100| ISS-JPEG10 | **29.07** | 67.48 | 63.82 |
> > |CIFAR-100| Ueraser | **25.58** | 72.08 | 55.65 |
> >
> > ImageNet-Subset:
> >
> > | | ISS-Gray | ISS-JPEG10 | Ueraser |
> > |:-:| :--- | :---: | :---: |
> > |Ours| 24.12 | 12.21 | 10.35 |
> >
> > As shown, our method provides superior protection against existing defense methods and outperforms other techniques. ISS [2] and Ueraser [3] have already demonstrated that other methods are highly vulnerable to these defenses.
> >
> > Regarding AVATAR:
> > 1. AVATAR implicitly assumes that the victim model has an estimator for the original data distribution. However, this assumption does not hold true in practice. As a victim model, it can only access unlearnable data.
> > 2. The code for AVATAR has not been made open-source, and the paper lacks specific parameter settings, making it difficult to reproduce. Once it becomes open-source, we will update the test results on arXiv accordingly.
> > 3. The idea of AVATAR comes from adversarial attacks [5]. However, this method has been broken by [6], which only requires adding constraints deviating from the true method during the denoise process of DDPM to defeat the method in [5]. This implies that if our $s_{\theta}$ induces samples towards a new data distribution direction, then AVATAR becomes ineffective.
> > We hope these additional results and clarifications address your concerns and demonstrate the effectiveness of our approach against various defense methods.
> >
> > > W5 : Please also include transformer-based models in Table 2.
> >
> > Thank you for your suggestion. We have modified Table 2. Our method still performs more superior than other methods.
> >
> > > Q1:  Are there significant advantages of the proposed method over OPS?
> >
> > We appreciate your inquiry about the significant advantages of our proposed method over OPS. We would like to highlight the following key benefits:
> > 1. In practice, **data augmentation methods are often employed** during the model training process, and their cost is **significantly lower** than that of adversarial training. As demonstrated in **W2**, while OPS can be easily compromised by data augmentation, our method is more robust against such techniques.  As demonstrated in Section **W4**, our method can resist defenses such as ISS and Ueraser, whereas OPS is vulnerable to them.
> > 2. The search space of OPS equals $H*W*C*255^3$, requiring a search through numerous possibilities for optimal results. As the image size increases, the performance of OPS is likely to degrade. In contrast, our method demonstrates effectiveness across datasets with varying resolutions and categories, as shown in Table 1.
> > 3. When the number of categories is large, the noise generated by OPS's search space may not satisfy linear separability, leading to a significant performance decline. Our method is not dependent on linear separability and is based on loss minimization, avoiding this issue and ensuring better stability.
> > 4. In OPS, the data of each category has the same noise at the same location, resulting in low concealment. Our method, on the other hand, targets specific samples with a smaller perturbation range, offering higher concealment.

---

> > > ### Author Response · Authors · 2023-11-23
> > >
> > > > Q2 : The proposed method seems to be only a minor modification on REM and requires even more computational overhead than REM.
> > >
> > > Thank you for your question. There might be some misunderstandings, and I will provide  explanations from two perspectives.
> > > **Method** : Our method is quite different from REM.
> > > 1. To enhance the transferability of unlearnable examples, we propose **Data Collapse**. By employing data collapse, we reduce the information contained in the data. The ablation experiments in Table 3 of the paper demonstrate that data collapse can indeed improve transferability, which is a feature not present in other related works.
> > > 2. To enhance the robustness of transferability, we propose Modified Adversarial Training (MAT). This method specifically modifies traditional adversarial training to address the unique characteristics of unlearnable examples.  The optimization objective of MAT is **(min-max)-min**, while the optimization objective of REM is **min-(min-max)**. **The two objectives are fundamentally different**. Our optimization objective aims to make the substitute model $f_{\theta}'$ robust, thereby making the generated unlearnable examples robust as well. In contrast, REM's goal is to generate unlearnable examples for adversarial samples, and its substitute model is not robust.
> > > 3. Our method achieves robust transferability through two steps: Data Collapse and Modified Adversarial Training.
> > >
> > > **Time**
> > > 1. Since the original data remains unchanged, training  requires only one iteration. This process takes approximately 8 hours of 1 V100 GPU time on CIFAR.
> > > 2. In our paper, we specifically employ the score matching method to induce data collapse. DDPM is essentially consistent with score matching, allowing us to utilize open-source checkpoints.
> > > 3. The table below presents a comparison of the overhead time between our method and existing methods such as EM, TAP, and REM:
> > >
> > > |Dataset|EM|TAP|REM|Ours|
> > > |:-|:-:|:-:|:-:|:-:|
> > > |CIFAR-10|0.4h|0.5h|22.6h| 8.6h |
> > > |CIFAR-100|0.4h|0.5h|22.6h| 8.6h |
> > > |ImageNet(subset)|3.9h|5.2h|51.2h| 26.5h|
> > >
> > >
> > >
> > > > Q3: Is the effectiveness of the proposed method still guaranteed if higher budget are considered for adversarial training? Or is it like REM where the unlearnable effect is significantly reduced?
> > >
> > > Thank you for your insightful question. As the adversarial training budget increases, the protective effect of unlearnable examples does decline.  This is a common limitation of unlearnable examples. The focus of our work is not on this aspect. Instead, we pay more attention to the issue of transferability.
> > >
> > > In response to your concern, we have increased both $\rho_u$ and $\rho_a$, and the experimental results are presented in the table below.
> > >
> > > | | $\rho_a = 0$ | $\rho_a=8/255$ | $\rho_a=16/255$ | $\rho_a=32/255$ | $\rho_a=64/255$ |
> > > | - | :-: | :-: | :-: | :-: | :-: |
> > > |Clean| 94.66 | 84.79 | 70.41 | 42.76 | 10.00
> > > |Ours($\rho_u=16/255$)| 10.21 | 15.80 | 71.73 | 43.75 | 10.00 |
> > > |Ours($\rho_u=32/255$) | 10.35 | 10.46 | 18.07 | 53.83 | 10.00 |
> > > |Ours($\rho_u=64/255$) | 10.15 | 10.27 | 11.01 | 54.30 | 10.00 |
> > >
> > > As you pointed out, increasing $\rho_a$$ does indeed reduce the effectiveness of unlearnable examples. To clarify, we define the party generating unlearnable examples as the Attacker and the party training models using unlearnable examples as the Defender. In classification tasks, the Attacker's goal is to minimize the test accuracy of the Defender's model trained on unlearnable examples, while the Defender's goal is to achieve higher model accuracy.
> > > 1. From the Attacker's perspective, increasing $\rho_u$ has little cost, as there is no need to use unlearnable examples. However, for the Defender, undermining the effect of unlearnable examples requires increasing $\rho_a$, which is costly. As seen in the table above, as $\rho_a$ increases, the performance upper bound of the model declines significantly. This is actually an arms race, where the Attacker forces the Defender to raise $\rho_a$ by increasing $\rho_u$. Once $\rho_a$ reaches a certain level, the model's training difficulty increases, and the performance upper bound is significantly reduced. In this way, the Attacker achieves their goal.
> > > 2. In this scenario, Table 1 in the paper indicates that our method is superior to other methods. Our approach provides a superior algorithm under these circumstances.
> > >
> > > > Other 1:  The design of the log-gradient estimator is unexplained.
> > >
> > > Thank you for pointing out the lack of explanation regarding the design of the log-gradient estimator. In our paper, we use , which employs the widely-used U-Net architecture. The score function is commonly used to estimate the direction of the distribution, as defined in [6]. We have revised our manuscript to include this information and provide a clearer understanding of the log-gradient estimator's design.

---

> > > > ### Author Response · Authors · 2023-11-23
> > > >
> > > > > Other 2:  Strangely, the hyperparameters $\rho_d$, $\alpha_d$ and $K_d$ are not defined properly, and $\rho_d$ is never assigned value or used in the paper. Also, the update on $x$ is unconstrained.
> > > >
> > > > Thank you for your careful observation regarding the hyperparameters. We apologize for any confusion caused by their lack of proper definition. In our paper, $\rho_d$  denotes the data collapse radius,  $\alpha_d$ represents the step size of data collapse, and  $K_d$ refers to the steps of data collapse. We have revised our manuscript to clearly define these hyperparameters in accordance with your feedback.
> > > >
> > > >
> > > > > Other3: Citations are not properly parenthesized, i.e. `\cite{…}` -> `\citep{…}`.
> > > >
> > > > Thank you for pointing out the improper formatting of citations in our manuscript. We have revised the paper and corrected the citation format as per your suggestion.
> > > >
> > > >
> > > > - \[1\] One-pixel shortcut: on the learning preference of deep neural networks. ICLR 2023
> > > > - \[2\] Image shortcut squeezing: Countering perturbative availability poisons with compression. ICML 2023
> > > > - \[3\] Learning the unlearnable: Adversarial augmentations suppress unlearnable example attacks. [https://arxiv.org/abs/2303.15127](https://arxiv.org/abs/2303.15127)
> > > > - \[4\] The Devil's Advocate: Shattering the Illusion of Unexploitable Data using Diffusion Models. [https://arxiv.org/abs/2303.08500](https://arxiv.org/abs/2303.08500)
> > > > - \[5\] [Guided diffusion model for adversarial purification](https://scholar.google.com/scholar?cluster=4953081698844866248&hl=zh-CN&as_sdt=2005&sciodt=0,5). ICML2022
> > > > - \[6\] DiffAttack: Evasion Attacks Against Diffusion-Based Adversarial Purification. NeurIPS2023
> > > > - \[7\] A Kernelized Stein Discrepancy for Goodness-of-fit Tests. ICML2016

---

> ### Author Response · Authors · 2023-11-23
>
> We are truly grateful for your valuable insights and comprehensive feedback. In our responses, we have endeavored to address every concern raised with thorough explanations and evidence. As we approach the conclusion of this phase of the review process, we are keen to ascertain whether our rebuttal has successfully resolved the issues highlighted.
> We invite any additional comments or questions you may have regarding our responses. Your expertise and perspectives are vital in guiding the refinement and understanding of our work. We eagerly await your further guidance and insight.

---

### Official Review · Reviewer_qwUi · 2023-11-03

**Soundness:** 3 good
**Presentation:** 3 good
**Contribution:** 3 good
**Rating:** 6
**Confidence:** 4

**Summary:**

The authors investigate the problem of generating robust transferable unlearnable data that can protect the data information against adversarial training. An algorithm based on data collapse is proposed and evaluated in numerical experiments using CIFAR-10 and CIFAR-100.

**Strengths:**

1. The mechanism of the proposed algorithm to protect the data from adversarial training is interesting.

2. The numerical results are impressive in general.

**Weaknesses:**

1. As pointed out by the authors in the last section, the proposed method requires significant computational cost and is currently unscalable to large dataset like ImageNet.

2. The proposed Algorithm 1 has many parameters. Can you provide more discussion on how to set them in practice based on data, and how are they related to the performance or robustness against adversarial training?

While I do not find other major weakness at this point, I do have additional questions in the next section.

**Questions:**

1. In Section 3.3, the predefined distribution is set as the normal distribution $N(\tilde{x};x, \sigma^2I)$. I am wondering what is the reason or intuition for such setting. Can we use a different distribution?

2. In Algorithm 1, the PGD process in lines 13-16 and lines 17-20 are quite similar. What is the advantage of two-stage PGD over one-stage? Also, is there any trade-off between the magnitude of $\rho_u$ and $\rho_a$?

3. Following the previous question, if the two-stage PGD is more powerful than one-stage, can we modify the algorithm to add more stages in PGD to expect a better performance?

4. In Table 1, while the proposed method outperforms the compared methods in most cases, it is less appealing than EM over larger dataset (CIFAR-100 and ImageNet Subset) when there is no adversarial training. Can you discuss the possible reason for that?

---

> ### Author Response · Authors · 2023-11-17
>
> Thanks for your valuable comments and your appreciation of our findings.
> To clarify, we did not merely compare CIFAR-10 and CIFAR-100; we also compared a subset of the first 100 classes of ImageNet. This covers data with different categories and resolutions. Subsequent transferability experiments were conducted only on CIFAR due to resource constraints.
> We address your concerns below:
>
> > W1. As pointed out by the authors in the last section, the proposed method requires significant computational cost and is currently unscalable to large dataset like ImageNet.
>
> Thank you for your comment. Detailly, in our setting, we employed 4 V100 GPUs to train ResNet-18 on the ImageNet-Subset for 5,000 steps, which takes only 39 hours. In the future, we will further explore how to improve the speed of our methods.
>
> > W2. The proposed Algorithm 1 has many parameters. Can you provide more discussion on how to set them in practice based on data, and how are they related to the performance or robustness against adversarial training?
>
>
> Thank you for your question.  For the data collapse step,
> $\alpha_d$ is based on commonly used default parameters from some score-based generation methods. As for $K_{\alpha}$, it is a hyperparameter, and its value is determined based on computational cost and the value of $\rho_u$.  For the Adv step, the step sizes $\alpha_a, \alpha_u$ are derived from the step counts $K_a, K_u$ and the perturbation ranges $\rho_a, \rho_u$. We usually ensure that $K_a \cdot \alpha_a > \rho_a, K_u \cdot \alpha_u > \rho_u$.
>
> The larger the $\rho_u$, the better the protection effect. The larger the $\rho_a$, the greater the range of adversarial training perturbations that can be protected, but it must be ensured that $\rho_a < \rho_u$. When the dataset resolution is large, $K_a, K_u$ typically choose smaller steps to reduce computational cost, and under constant $\rho_a, \rho_u$, $\alpha_a, \alpha_u$ need to be larger. Generally speaking, the more steps $K_a, K_u$ have, the better the effect.
>
>
> > Q1. In Section 3.3, the predefined distribution is set as the normal distribution $\mathcal{N}(\hat{x};x,\sigma^2I)$. I am wondering what is the reason or intuition for such setting. Can we use a different distribution?
>
> Thank you for your insightful question.
> Firstly, our training objective is to estimate the true data distribution $p_D(x)$, but we do not have the ground truth of the real data. Secondly, according to Stochastic Gradient Langevin Dynamics, what we actually need is $\nabla_x \log p_D{(x)}$, which is referred to as the Score, with the full name being Stein Score. Therefore, we estimate the Score using the denoise score matching [1] method.
>
> In the Denoise Score Matching process, we add Gaussian noise $\epsilon = \mathcal{N}(0, \sigma^2)$, resulting in $\tilde{x} = x + \epsilon$. According to the properties of the Gaussian distribution, we obtain the distribution $q(\tilde{x}|x)= \mathcal{N}(\tilde{x};x,\sigma^2I)$ as mentioned in the paper.
> The optimization objective of Denoise Score Matching is Equation 3:  $$\frac{1}{2} \mathbb{E}_{q_\sigma(\tilde{x} \mid x) p_D(x)}\left[\left\|s_\theta(\tilde{x})-\nabla_{\tilde{x}} \log q_{\sigma}(\tilde{x}|x)\right\|_2^2\right]$$Therefore, to make the equation meanfuling, any distribution that satisfies the existence of $\nabla_{\tilde{x}} \log q_{\sigma}(\tilde{x}|x)$ can be used.
>
>
>
> > Q2. In Algorithm 1, the PGD process in lines 13-16 and lines 17-20 are quite similar. What is the advantage of two-stage PGD over one-stage? Also, is there any trade-off between the magnitude of $\rho_a$ and $\rho_u$?
>
> Thank you for your question.  PGD[2] is a standard approach for solving inner maximization and minimization problems.  It performs iterative projection updates to search for the optimal perturbation.
>
> In our modified adversarial training, the optimization formula is min-max-min. Our algorithm incorporates two PGD processes. The first PGD process corresponds to the inner min in the optimization formula, solving for $\delta_u$. The second PGD process corresponds to the max procedure in the optimization formula, solving for $\delta_a$. The larger the $\rho_u$, the better the protection effect. The larger the $\rho_a$, the greater the range of adversarial training perturbations that can be protected, but it must be ensured that $\rho_a < \rho_u$.
>
> > Q3. Following the previous question, if the two-stage PGD is more powerful than one-stage, can we modify the algorithm to add more stages in PGD to expect a better performance?
>
> Thank you for your question. As mentioned in Q2, the PGD step is used to different function. Each PGD step corresponds to an optimization objective, so it cannot be arbitrarily added.

---

> > ### Author Response · Authors · 2023-11-17
> >
> > > Q4. In Table 1, while the proposed method outperforms the compared methods in most cases, it is less appealing than EM over larger dataset (CIFAR-100 and ImageNet Subset) when there is no adversarial training. Can you discuss the possible reason for that?
> >
> > Thank you for your insightful question.
> > Firstly, in Table 1, both the substitute model and the poisoned model used are ResNet-18. If the poisoned model does not undergo adversarial training, it is non-robust.
> > The optimization objective of EM is :
> > $$\min_{\theta} \frac{1}{n}\sum_{i=1}^n \min_{||\delta_i||\leq \rho_u}\ell(f_{\theta}(x_i+\delta_i), y_i)$$
> > Evidently, the alternative model for EM is non-robust, akin to a white-box attack. In contrast, our substitute model, after undergoing modified adversarial training, exhibits robustness. This deviates significantly from the training methodology employed by the poisoned model, rendering its performance inferior to that of EM, which is to be expected. Such a phenomenon is commonly observed in both standardly trained and adversarially trained models, essentially representing a trade-off between accuracy and robustness.
> >
> > Moreover, in Table 3, w/o (Collapse + Adv) essentially refers to the EM algorithm, while w/o (Adv) denotes the absence of adversarial training, with only data collapse being performed. It can be observed that the protective effect on ResNet-18 is consistent with that of EM, while the transferability exhibits a substantial improvement relative to EM.
> >
> > - \[1\] A connection between score matching and denoising autoencoders. Neural computation 2011
> > - \[2\] Towards deep learning models resistant to adversarial attack. ICLR 2018

---

> ### Comment · Reviewer_qwUi · 2023-11-17
>
> Thanks much for the response and clarifications. I misunderstood a small part of the algorithm in the beginning. I think my previous questions are addressed properly.
>
> As I go through other reviews, I find another major concern that I did not think of earlier, as raised by reviewer KXx5.
>
> In the numerical studies, for adversarial training it seems only the approach in [1] is considered. I am curious about the performance of the work with adversarial training (or other robust) schemes proposed more recently.
>
> [1] Aleksander Madry, Aleksandar Makelov, Ludwig Schmidt, Dimitris Tsipras, and Adrian Vladu.
> Towards deep learning models resistant to adversarial attacks. In 6th International Conference on
> Learning Representations, ICLR 2018, Vancouver, BC, Canada, April 30 - May 3, 2018, Conference
> Track Proceedings. OpenReview.net, 2018. URL https://openreview.net/forum?id=
> rJzIBfZAb.

---

> > ### Author Response · Authors · 2023-11-23
> >
> > > In the numerical studies, for adversarial training it seems only the approach in [1] is considered. I am curious about the performance of the work with adversarial training (or other robust) schemes proposed more recently.
> >
> > Thank you for your insightful response. The optimization objective for adversarial training is $$\min_{\theta} \frac{1}{n}\sum_{i=1}^n \max_{||\delta_i||\leq \rho_a}\ell(f_{\theta}(x_i+\delta_i), y_i)$$
> > Although existing methods are based on this objective, they all incorporate minor modifications. To more comprehensively evaluate the robustness of our method, we conducted experiments on recent adversarial training methods, and the results are as follows.
> >
> > | Dataset | PGD-10 | FGSM-CS[1] | IDBH[2] |
> > | :-: | :-: | :-: | :-:|
> > |CIFAR-10| 25.77 | 16.77 | 20.33 |
> >
> > As demonstrated in the above table, even recent work([1,2]) still can't destroy the effect of unlearnable examples.
> >
> > - \[1\] Fast Adversarial Training with Smooth Convergence. ICCV2023
> > - \[2\] Data Augmentation Alone Can Improve Adversarial Training. ICLR 2023

---

> ### Author Response · Authors · 2023-11-23
>
> We are truly grateful for your valuable insights and comprehensive feedback. In our responses, we have endeavored to address every concern raised with thorough explanations and evidence. As we approach the conclusion of this phase of the review process, we are keen to ascertain whether our rebuttal has successfully resolved the issues highlighted.
> We invite any additional comments or questions you may have regarding our responses. Your expertise and perspectives are vital in guiding the refinement and understanding of our work. We eagerly await your further guidance and insight.

---

### Official Review · Reviewer_czgC · 2023-11-06

**Soundness:** 2 fair
**Presentation:** 2 fair
**Contribution:** 2 fair
**Rating:** 5
**Confidence:** 3

**Summary:**

This paper is concerned with approaches that can help protect data against unauthorised use by learning algorithms by crafting a noise that makes the data “unlearnable”. They propose to first train a log gradient estimator offline which is used to modify the training samples with an SGLD-like update. This is followed by incorporating PGD-noise over an adversarially trained surrogate model. Extensive empirical evaluations are performed to test the transferability of the proposed method and compare it across two methods.

**Strengths:**

The work is novel in its formulation of transforming training samples to make them unlearnable via a two-step process - a model-independent modification of data samples to reduce the variability in the the data distribution, followed by incorporation of PGD-noise over an adversarially trained surrogate.

**Weaknesses:**

- While transferability has been emphasised as a distinction, I think its interpretation relative to Robust Unlearning Methods needs to be clearly defined.

- From a presentation and readability standpoint, I think including the arguments for “how” this approach tackles transferability upfront would better clarify the contributions. Moreover, it might be worth considering if the aim is to ensure universality rather than transferability.

**Questions:**

- Does the sequence of Collapse and Adv matter?

- It would be useful to clearly describe the second step in term of what f represents and how PGD over adversarially trained model is carried out. I think the algorithm has the required steps but the description in Section 3.4 wasn’t sufficiently clear.

- In the current notations model f and model s are defined with shared parameters of \theta. Please clarify

- From my understanding the universailty/transferability relies on the first step of modification of data distribution. This approach to model-independent modification needs to chould be positioned in contrast to related works. For instance, can this be combined with other approaches as a pre/post processing step?

---

> ### Author Response · Authors · 2023-11-17
>
> Thanks for your valuable comments and your appreciation of our findings. We address your concerns below:
>
> > W1. The interpretation relative of transferability to Robust Unlearning Methods.
>
> Thank you for your suggestions.  We have revised our manuscript to clarify this point. Our transferability refers to the effectiveness of unlearnable examples generated by different surrogate models on various poisoned models, regardless of whether they employ standard training or adversarial training strategies. We have highlight the definition in the introduction with italic type.
>
> > W2. From a presentation and readability standpoint, I think including the arguments for “how” this approach tackles transferability upfront would better clarify the contributions. Moreover, it might be worth considering if the aim is to ensure universality rather than transferability.
>
> Thank you for your suggestion. We have summarized the arguments and put them upfront our approach in Section 3.3.  $x_t$ represents the sample at step $t$; $\nabla \log p_D(x_t)$ denotes the $\log$ gradient of the data distribution concerning the sample $x_t$; $\alpha$ signifies the step size; $\epsilon$ represents random noise following a distribution. $s_{\theta}$ indicates the data distribution gradient estimator, while $\tilde{x}$ refers to the sample after adding noise.  Universality is also an important research direction, and we will further deepen our research in this field in the future.
>
> > Q1. Does the sequence of Collapse and Adv matter?
>
> Thank you for your  insightful question regarding the sequence of Collapse and Adv. Yes, the order in which they are performed does matter. We first apply Collapse to modify the data, aiming to enhance the transferability of unlearnable examples. Subsequently, we conduct the modified Adv to improve the robustness of transferability.
> If we were to perform Adv first, the resulting noise would be model-dependent, and the subsequent Collapse would be based on this model-dependent foundation, significantly reducing transferability. Moreover, if the noise generated by Adv varies each time, a new data distribution estimator would need to be trained for each instance. By executing Collapse before Adv, we avoid this issue, as Collapse targets the original data.
>
> > Q2. Clearly describe the the second step in term of what f represents and how PGD over adversarially trained model is carried out.<It would be useful to clearly describe the second step in term of what f represents and how PGD over adversarially trained model is carried out. I think the algorithm has the required steps but the description in Section 3.4 wasn’t sufficiently clear.
>
> Thank you for your question.
> 1. $f_{\theta}'$ represents the surrogate model, which is distinct from the $s_{\theta}$ used in the first step for inducing data exploration. These are two separate models.
> 2. PGD[1] is a standard approach for solving inner maximization and minimization problems. It performs iterative projection updates to search for the optimal perturbation as follows:
> $$\delta^{(k)} = \prod_{\| \delta \| \leq \rho} \left[ \delta^{(k-1)} + c \cdot \alpha \cdot \text{sign} \left( \frac{\partial}{\partial \delta} \ell(f_\theta(x + \delta^{(k-1)}), y) \right) \right] $$
> 	Where $k$ is the current iteration step ($K$ steps at all), $\delta^{(k)}$ is the perturbation found in the $k$-th iteration, $c\in\{-1,1\}$ is a factor for controlling the gradient direction, $\alpha$ is the step size, and $\prod_{\|\delta\|\leq\rho}$ means the projection is calculated in the ball sphere $\{ \delta:\|\delta\| \leq \rho \}$. The final output perturbation is $\delta^{(K)}$. Throughout this paper, the coefficient $c$ is set as $1$ when solving maximization problems and $-1$ when solving minimization problems. This also can be found in Appendix A.3.
> 3. We employ modified adversarial training to update the surrogate model in three main steps. First, we use the PGD algorithm to generate unlearnable noise for clean samples, obtaining $x+\delta^u$. Second, we apply the PGD algorithm to $x+\delta^u$ to produce adversarial noise, resulting in $x+\delta^u+\delta^a$. Finally, we input the obtained $x+\delta^u+\delta^a$ into the surrogate model and update the surrogate model parameters using stochastic gradient descent. We have modified the Section3.4.
>
> > Q3. Is the current notations model f and model s are defined with shared parameters of $\theta$. Please clarify
>
> Thank you for your question. No, the current notations for the models $f_{\theta}'$ and $s_{\theta}$ represent different parameters. For $s_{\theta}$, it represents the log gradient of the data distribution estimator, and we use the U-Net structure. While for $f'_{\theta}$, it represents the surrogate model for generating robust unlearnable examples, and we employ common architectures such as VGG-16, ResNet-18, ResNet-50, DenseNet-121, and WRN-34-10.
>
> - \[1\] Towards deep learning models resistant to adversarial attacks. ICLR2018

---

> ### Author Response · Authors · 2023-11-17
>
> > Q4. From my understanding the universailty/transferability relies on the first step of modification of data distribution. This approach to model-independent modification needs to should be positioned in contrast to related works. For instance, can this be combined with other approaches as a pre/post processing step?
>
> Thank you for your insightful question.  Yes, the data collaspe step can be combined with other related methods. The following table shows the results on CIFAR10.
>
> | | VGG-16 | ResNet-18 | ResNet-50 | DenseNet-121 | WRN-34-10| ViT |
> | :- | :-: | :-: | :-: | :-: | :-: | :-: |
> | EM | 15.70 | 13.20 | **10.19** | 14.59 | 10.56 | 72.62 |
> | EM+(**Data Collapse**) | **11.20** | **11.44** | 10.72 | **12.50** | **10.23** | **19.02** |
> | TAP | 29.39 | 22.15 | 27.69 | 81.63 | 27.04| 52.28 |
> | TAP+(**Data Collaspe**) | **23.85** | **15.57** | **22.87** | **75.78** |  **20.89** | **47.15** |
>
>
> Table 3 in the paper also demonstrates that our data collapse step can be combined with other methods. In Table 3 of the paper, **w/o(Collaspe + Adv) = EM**, **w/o(Adv) = EM + Data Collapse**. Both Table 3 and the table above show that our data collapse step can be integrated with other methods and can improve transferability.

---

> ### Author Response · Authors · 2023-11-23
>
> We are truly grateful for your valuable insights and comprehensive feedback. In our responses, we have endeavored to address every concern raised with thorough explanations and evidence. As we approach the conclusion of this phase of the review process, we are keen to ascertain whether our rebuttal has successfully resolved the issues highlighted.
> We invite any additional comments or questions you may have regarding our responses. Your expertise and perspectives are vital in guiding the refinement and understanding of our work. We eagerly await your further guidance and insight.

---

### Meta-Review · Area_Chair_wTPc · 2023-12-11

**Metareview:**

This paper investigates methods for defending data from unauthorized use by learning algorithms by generating noise that renders the data “unlearnable.” The authors propose first training an offline log gradient estimator, which is then used to modify the training samples with an SGLD-like update. This is followed by incorporating PGD-noise over an adversarially trained surrogate model. Extensive empirical evaluations are conducted to assess the transferability of the proposed method and compare it to two other methods.

Strengths: The authors claim that the paper is the first to apply out-of-distribution perturbation to unlearnable examples. Experiments show that the proposed method generates unlearnable examples with superior robust transferability.

Weaknesses: I concur with the concerns raised by Reviewer KXx5 and Reviewer qwUi regarding the computational cost. The min-max optimization problem, which is required at each iteration, appears to be computationally expensive to solve. This may significantly restrict the algorithm's usability. In addition, the term "learnability" has a clear definition in the framework of PAC learnability in statistical learning theory. First, the authors should identify some connections with PAC learnability. Second, as an additional suggestion, this paper would benefit greatly from including a theory for a very simple machine learning model, such as linear regression, and demonstrating that this method indeed increases the test risk. I believe that a theory can be obtained for linear regression. This will make this paper more convincing.

**Justification For Why Not Higher Score:**

See the weaknesses above.

**Justification For Why Not Lower Score:**

N/A

---

### Decision · Program_Chairs · 2024-01-16

Reject